# Activation and inhibition of the C-terminal kinase domain of p90 ribosomal S6 kinases

Marlene Uglebjerg Fruergaard[1], Christine Juul Fælled Nielsen[1], Cecilia Rosada Kjeldsen[2], Lars Iversen[2], Jacob Lauwring Andersen[3], Poul Nissen[1]

The p90 ribosomal S6 kinases (RSKs) contain two distinct catalytic kinase domains, the N-terminal and C-terminal kinase domains (NTKD and CTKD, respectively). The activation of CTKD is regulated by phosphorylation by extracellular signal–regulated kinase (ERK1/2) and an autoinhibitory αL helix. Through a mutational series in vitro of the RSK CTKDs, we found a complex mechanism lifting autoinhibition that led us to design constitutively active RSK CTKDs. These are based on a phosphomimetic mutation and a C-terminal truncation (e.g., RSK2 T577E D694*) where a high activity in absence of ERK phosphorylation is obtained. Using these constructs, we characterize $IC_{50}$ values of ATP-competitive inhibitors and provide a setup for determining specificity constants ($k_{inact}/K_i$) of covalent CTKD inhibitors.

## Introduction

The p90 ribosomal S6 kinases (RSKs) are a group of highly related serine/threonine kinases operating downstream of the Ras/Raf/MAPK (mitogen-activated protein kinase) pathway [1]. The kinase family consists of four human isoforms (RSK1-4) and two mitogen- and stress-activated kinases (MSK1-2). The RSK/MSKs contain a N-terminal kinase domain (NTKD) sharing homology with the protein kinase A/G/C family, a conserved linker region, and a C-terminal kinase domain (CTKD) belonging to the $Ca^{2+}$/calmodulin–dependent protein kinases (CaMKs). RSK1-3 are ubiquitously expressed in every human tissue and regulate many cellular processes, such as cell growth, proliferation, and metabolism [2, 3]. Up-regulation of the RSK1/2 has been linked to promote cancer cell growth (e.g., breast cancer [4] and prostate cancer [5]), whereas in contrast, RSK3 and 4 show tumor suppressor functions [6, 7]. Heterogeneous loss-of-function mutations of the human RSK2 gene (RPS6KA3) have been implicated in Coffin–Lowry syndrome, which is characterized by multiple symptoms including

growth retardation and cognitive impairment [8]. Furthermore, the MAPK pathway is known to regulate differentiation of Th1 and Th17 cells (T helper cell 1 and 17), which are involved in the pathogenesis of autoimmune diseases as psoriasis and multiple sclerosis [9, 10, 11]. The RSKs therefore have emerged as promising therapeutic targets for several human diseases [7].

The RSK/MSK activity is regulated through a sequential phosphorylation cascade. The activation pathway is initiated by phosphorylation of the activation loop (T577 in human RSK2 numbering) in the CTKD by extracellular signal–regulated kinase 1 or 2 (ERK1/2 or p38 for the MSKs [1]), resulting in CTKD activation. Moreover, two additional sites in the linker region (T363 and S369) are phosphorylated by ERK1/2. Afterward, a hydrophobic motif within the linker region, S386 is autophosphorylated by the activated CTKD, creating a docking site for 3'-phosphoinositide–dependent kinase-1 (PDK1). The NTKD is phosphorylated by PDK1 (S227), leading to the fully activated RSK/MSKs that can phosphorylate substrates by the NTKD [3]. For RSK1/2, regulation by an autoinhibitory αL helix in the CTKD has been characterized (Fig 1). The αL helix occupies a "cradle" shaped by the αF–αG junction, thereby blocking for substrate binding. Docking of ERK1/2 at the C terminus is expected to promote αL displacement and CTKD activation [12, 13, 14].

Several classes of RSK inhibitors have been characterized. The kaempferol glycoside SL0101, extracted from the tropical plant *Forsteronia refracta*, was the first specific inhibitor of the RSK isoforms identified. SL0101 competes for ATP binding at the NTKD and was reported to inhibit RSK2 with an $IC_{50}$ of 89 nM [2, 3, 4]. The dihydropteridinone BI-D1870 is another selective NTKD ATP antagonist having an in vitro $IC_{50}$ of ~15–30 nM. Although both inhibitors have shown relative selectivity for the RSKs in a panel of purified kinases, several off-targets were observed at higher concentrations, including Aurora B and PIM3 [3, 7].

To our knowledge, selective reversible ATP-competitive inhibitors of the CTKD have not been reported. However, in a comprehensive analysis of kinase inhibitor selectivity by Davis et al [15] (72 known inhibitors screened against 442 kinases), a small subset of

[1]Department of Molecular Biology and Genetics, DANDRITE – Nordic EMBL Partnership for Molecular Medicine, Aarhus University, Aarhus C, Denmark   [2]Department of Clinical Medicine– The Department of Dermatology and Venereology, Aarhus N, Denmark   [3]Department of Biomedicine, Aarhus University, Aarhus C, Denmark

Correspondence: pn@mbg.au.dk
Cecilia Rosada Kjeldsen's present address is Skanderborg kommune, Denmark
Jacob Lauwring Andersen's present address is Meelunie GPI, Hedensted, Denmark

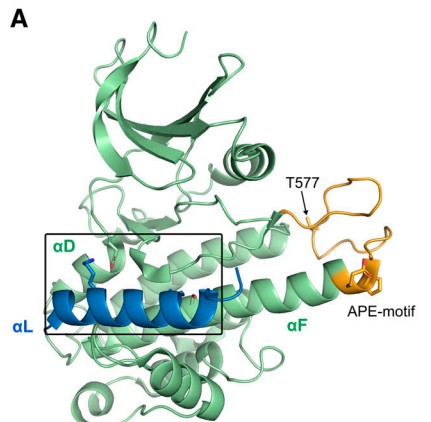

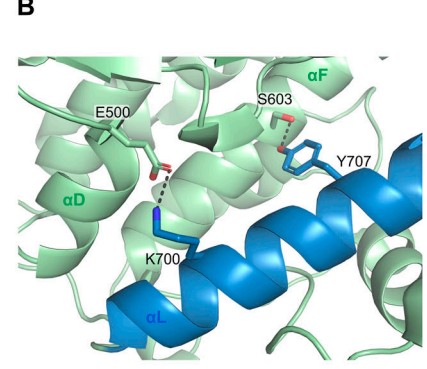

**Figure 1. Crystal structure of the inactive RSK2 C-terminal kinase domain.**
**(A)** In the inactive state of the RSK2 C-terminal kinase domain (PDB 2QR8 (12)), the autoinhibitory αL helix (blue) is stabilized through ionic and hydrogen bond interactions to αD and αF. Depicted in orange, the activation loop is illustrated in cartoon, and the Ala-Pro-Glu motif (APE motif, conserved segment of the kinase activation loop or the P loop involved in ATP cofactor binding) and T577 are shown as sticks. **(A, B)** Zoom of the boxed region in (A) illustrating the E500(αD)-K700(αL) and S603(αF)-Y707(αL) interactions.

compounds were identified displaying moderate specificity for the RSK CTKDs, including ruxolitinib (INCB18424), SB-203580, and TG-100-115 (15). The third inhibitor class consists of irreversible RSK CTKD inhibitors that specifically modify reactive cysteines. The potent fluoromethylketone pyrrolopyrimidine (fmk) inhibitor targets CTKD of RSK1, RSK2, and RSK4 and was shown to inhibit RSK2 with an $IC_{50}$ of 15 nM. The fluoromethylketone motif functions as the reactive electrophile that covalently binds Cys436 in the ATP-binding pocket of RSK2 CTKD (7, 16, 17). Dimethyl fumarate (DMF), used in the treatment of psoriasis and multiple sclerosis, was recently identified to primarily react as a Michael acceptor to another cysteine, a conserved C599 at an allosteric site of RSK2 CTKD (18). Binding of DMF to C599 prevents the phosphorylated activation loop to undergo structural rearrangement during activation, and thereby inhibiting the phosphorylated T577 from coordinating with a lysine of the substrate. DMF showed moderate efficacy of 225 $\mu$M against the inactive RSK2 CTKD (18).

To obtain a deeper understanding of the activation and regulation of the RSK CTKDs, we performed a series of mutational studies on the RSK2 CTKD. We found that the in vitro stabilization of the autoinhibitory helix in the inactive conformation is more complex than previously considered. These findings led to the generation of constitutively active mutants of the RSK CTKDs, which can be of potential use in screening campaigns for RSK mechanisms and CTKD kinase inhibitors, preventing the need for complex activation mechanisms that may obscure mechanistic interpretations.

## Results

The RSK CTKDs are activated by phosphorylation of conserved threonine residue in the activation loop (T577 in human RSK2 numbering) by the upstream kinase ERK1/2. In our attempt to generate a constitutively active RSK2 CTKD, we introduced a phosphomimetic T577E mutation. Furthermore, the autoinhibitory αL-helix is stabilized by a hydrogen bond between S603 (αF) and Y707 (αL), and its inhibitory role is strengthened by an ionic pair between E500 (αD) and K700 (αL) (12). Alanine point mutations of K700 and Y707 therefore were introduced to disrupt these interactions. Lastly, constructs with removal of αL (ΔαL) or CTKD

truncated at the N terminus of the αL helix (marked with asterisk) were made.

The activity of RSK2-T577E and other RSK CTKD contructs (see Fig 2A) were evaluated in a well-established, time-resolved fluorescence resonance energy transfer assay on kinase activity (18) (establishment of assay for RSK2 CTKD; see Fig 2B and C). In brief, an anti-phospho STK-substrate antibody labeled with $Eu^{3+}$ cryptate (FRET donor) recognizes only the phosphorylated biotinylated STK-substrate. Binding of streptavidin-XL665 (FRET acceptor) to the biotinylated substrate brings the donor and acceptor into close proximity and FRET signals are generated (19). Each construct was activated with ERK2 (+ERK) or activity was measured directly without preincubation (−ERK), to validate their ERK independent, that is, constitutive activity (comparing (+ERK) and (−ERK) in Fig 2D and E). The RSK2-T577E mutant displayed a low 8.4% gain of constitutive activity relative to fully activated WT (Fig 2D). In the following, point mutations of key residues of the αL helix (interfering with stabilizing interactions to neighbouring helices) were combined with the T577E mutation in order to further relieve the autoinhibition (K700A, Y707A, and K700A + Y707A) (12). All three mutants, that is, RSK2-T577E-K700A, RSK2-T577E-Y707A, and RSK2-T577E-K700A-Y707A showed similar (−ERK) activity levels of 5.4, 6.0, and 9.2% of wt RSK2 (+ERK) activity, that is, not different from the RSK2-T577E mutant alone. A mutant with an αL helix truncation, but with preserved C terminus (RSK2-T577E-ΔαL) and a mutant with C-terminal truncation before the αL helix (RSK2-T577E-D694*) were also created, and the latter displayed a marked constitutive activity (66.5%) of ERK-activated wt RSK2 activity. It is worth noting that the mutant harboring only the truncation (RSK2-D694*) displayed only significant activity upon ERK activation.

Analogous mutants of RSK2-T577E-D694* were generated also for RSK1 and RSK4 (RSK1-T573E-D690* and RSK4-T581E-D698*), and they also showed clear constitutive activity (Fig 2E). RSK3 construct was only expressed as inclusion bodies in *E. coli*, and further investigations of RSK3-T570E-D687* were not pursued, but it is expected to also show similar properties of constitutive activity.

Next, we investigated if the constitutive activity of the RSK2 CTKD would also manifest in a cellular model. Serum-starved HEK293 cells were transfected with full-length human RSK2 (wt) and human RSK2 with the activated CTKD (T577E-D694*). The cells were kept in a serum-free medium until termination (0, 6, 15, 24, and 48 h after

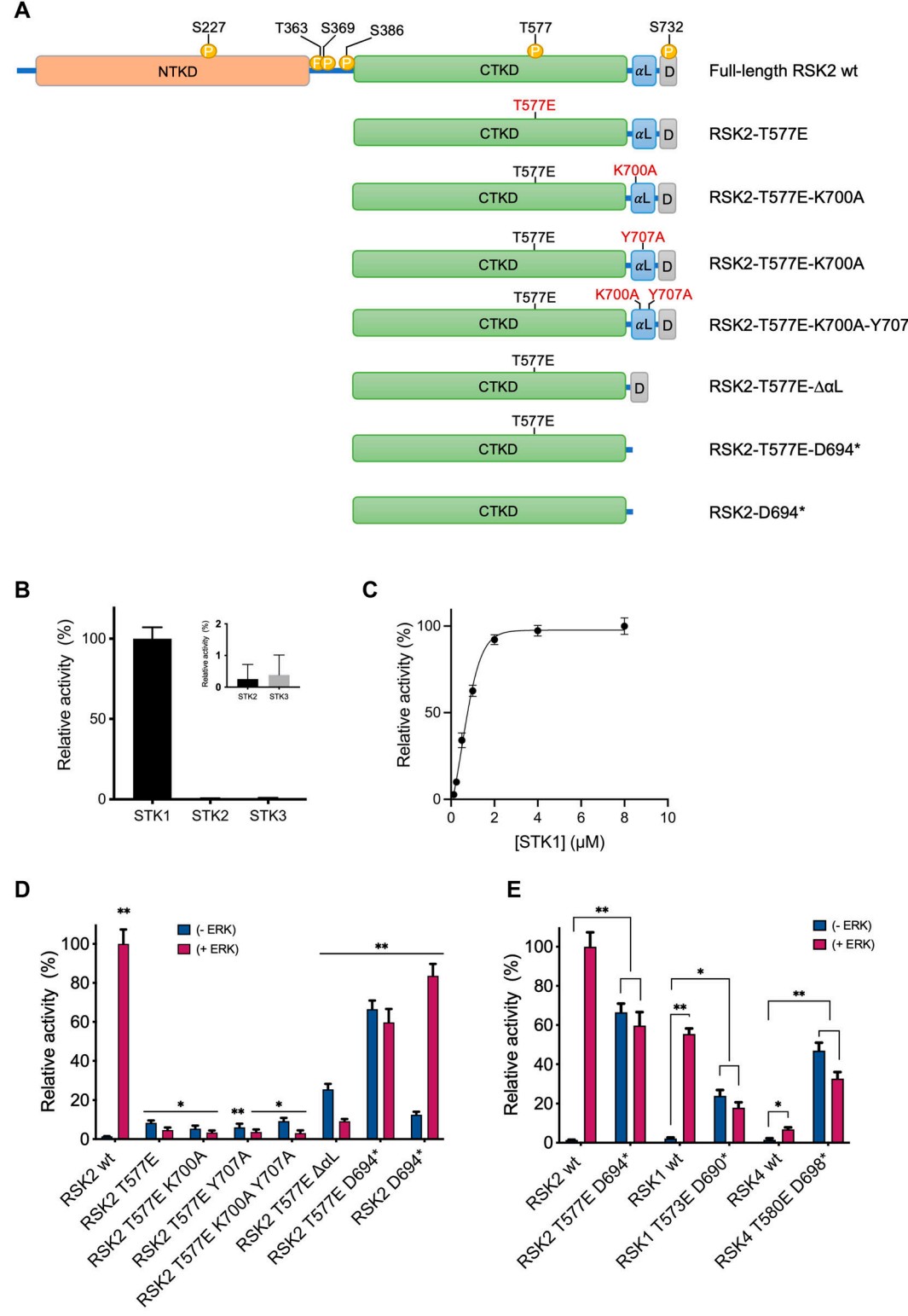

**Figure 2. RSK constructs and HTRF KinEASE assay establishment.**
**(A)** Schematic representation of full-length RSK2 wt, RSK2 CTKD wt, and RSK CTKD mutants with each newly introduced mutation marked in red. The domains are depicted using the following color code; orange (N-terminal kinase domain), green (CTKD), blue (αL-helix), grey (ERK docking motif [D domain]), and yellow (phosphorylation sites). Mutational studies in vitro were all performed on human RSK CTKDs, so for simplicity the CTKD abbreviation is not included in the construct names. **(B)** STK substrate screening. Activity of RSK2 CTKD wt with STK1 was set to 100% (n = 3). All error bars indicate mean ± SD. **(C)** STK1 titration resulting in $K_M$(STK1) of 0.6 ± 0.1 μM. **(D, E)** Relative activity of RSK CTKD mutants untreated or pre-treated with 1 h kinase activation with ERK2 (−ERK: blue and +ERK: red). The activity of RSK2 CTKD wt (+ERK) was set to 100% (n = 3–6). **(D, E)** Statistical significance was measured via unpaired and two-tailed *t* tests (*P < 0.01; **P < 0.0001; *t* test versus RSK2 wt (−ERK) in (D) or the related RSK wt (1/2/4) (−ERK) in (E), respectively). All error bars indicate mean ± SD.

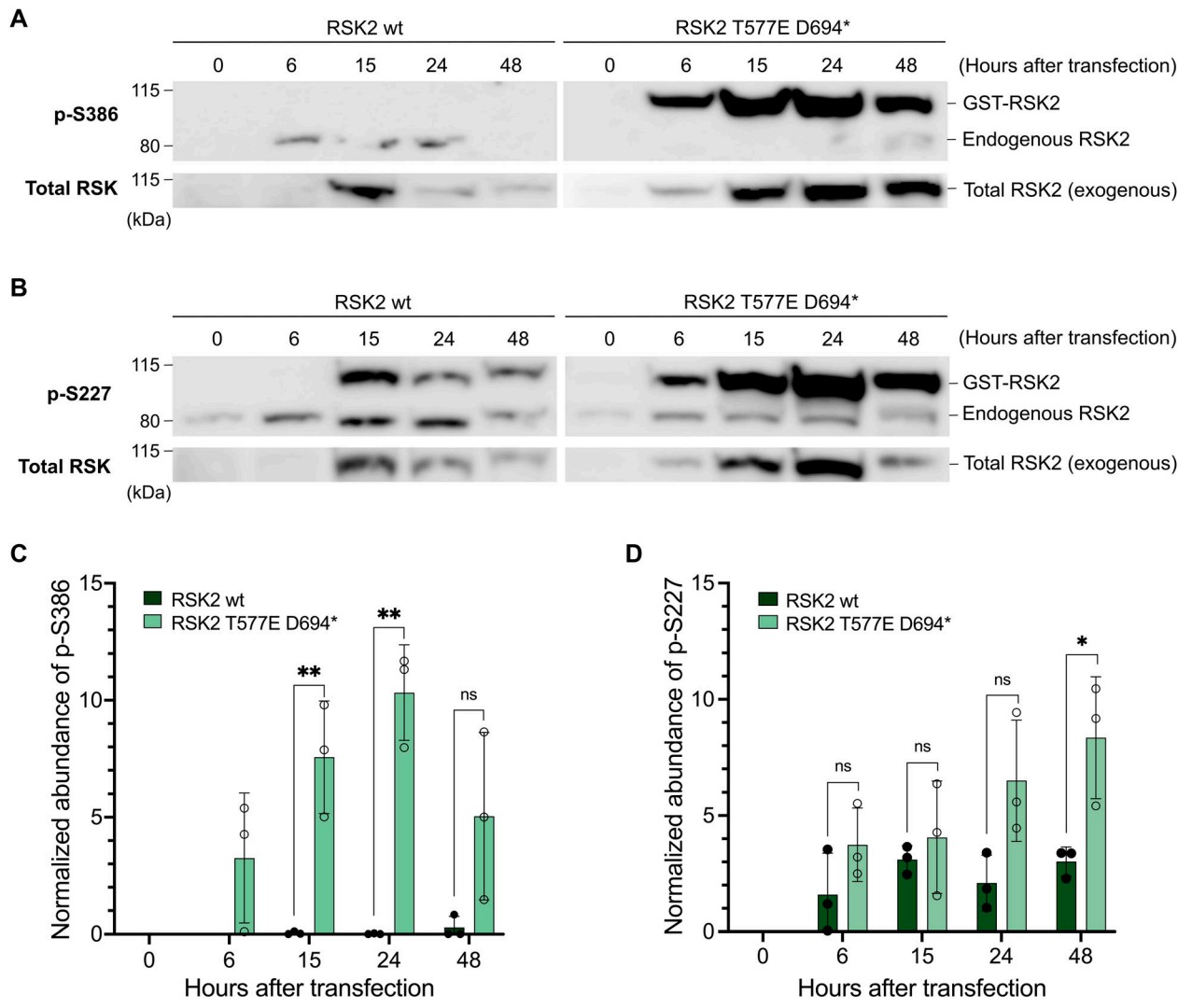

**Figure 3.  HEK293 cell study of full-length human RSK2 wt and "full-length" human RSK2 T577E D694* mutant.**
**(A, B)** Representative Western blots from RSK2 wt and RSK2 T577E D694* phosphorylation analysis of S386 (p-S386) and S227 (p-S227), respectively. **(C, D)** CTKD activity was determined as S386 phoshorylation and activated NTKD (D) was determined as S227 phosphorylation (RSK2 wt: dark green and RSK2-T577E-D694*: light green). The relative abundance of p-S386 and p-S227 were normalized to the total exogenous RSK2 levels (n = 3; all data points are indicated by circles). Statistical significance was measured via unpaired and two-tailed *t* tests (**P* < 0.05, and ***P* < 0.01). All error bars indicate mean ± SD.
Source data are available for this figure.

transfection) to prevent extracellular stimulus of the MAPK-signaling pathway (Fig 3). The activity of the kinase domains (CTKD and NTKD) was determined by Western blot analysis using phospho-specific antibodies. For the CTKD, autophosphorylation was detected by S386 phosphorylation (p-S386). To validate a presumed NTKD activity of the mutant, investigation of downstream phosphorylation was attempted for two RSK substrates; the transcription factor cAMP-response element binding protein (CREB) and transcription/translation factor Y-box binding protein-1 (YB-1) (6, 20). However, high background phosphorylation levels were observed for both substrates hindering a clear detection of RSK2-dependent phosphorylation (data not shown). Therefore, activated CTDK was probed by the PDK1-dependent phosphorylation of S227 in the NTKD of RSK (p-S227). Based on this readout, RSK2-T577E-D694* displayed a clear CTKD activation with nearly 2.8-fold higher phosphorylation of S227 observed for RSK2-T577E-D694* mutant (after 48 h) compared with wt RSK2 (Fig 3C and D).

Biochemical characterization of RSK/MSK kinase function is obscured by the numerous interactions and activation steps involved, therefore also implicating multiple ATP binding sites. Generation of constitutively active kinase mutants enables a direct biochemical characterisation and determination of apparent $K_M$ for ATP for the activated RSK CTKDs, as the interference with another ATP-dependent pre-phosphorylation procedure by ERK2 is not required. Using the constitutively active RSK mutant, the apparent Km for ATP, $K_M^{ATP}{}_{app}$ was determined for the three human RSK CTDK constructs (RSK1-T573E-D690*, RSK2-T577E-D694*, and RSK4-T581E-D698*) at 107 $\mu M$, 40 $\mu M$, and 87 $\mu M$, respectively (Table 1).

**Table 1.  Apparent ATP $K_M$ values.**

| $K_m$(app) | RSK1-T573E-D690* | RSK2-T577e-D694* | RSK4-T581E-D698* |
|---|---|---|---|
| ATP | 107 ± 17 μM | 40 ± 3 μM | 87 ± 7 μM |

**Table 2.  Table of ruxolitinib, SG-203580, TG-100-115, and fmk inhibition ($IC_{50}$ values) of RSK1-T573E-D690*, RSK2-T577E-D694*, and RSK4-T581E-D698*.**

| $IC_{50}$ | RSK1-T573E-D690* | RSK2-T577E-D694* | RSK4-T581E-D698* |
|---|---|---|---|
| Ruxolitinib | 0.7 ± 0.1 μM | 3.2 ± 0.4 μM | 2.6 ± 0.4 μM |
| SB-203580 | 1.9 ± 0.8 μM | 3.6 ± 0.6 μM | 4.6 ± 0.9 μM |
| TG-100-115 | 1.7 ± 0.4 μM | 33 ± 4 μM | 14 ± 2 μM |
| fmk | 63 ± 10 nM | 13.7 ± 0.9 nM | 61 ± 5 nM |

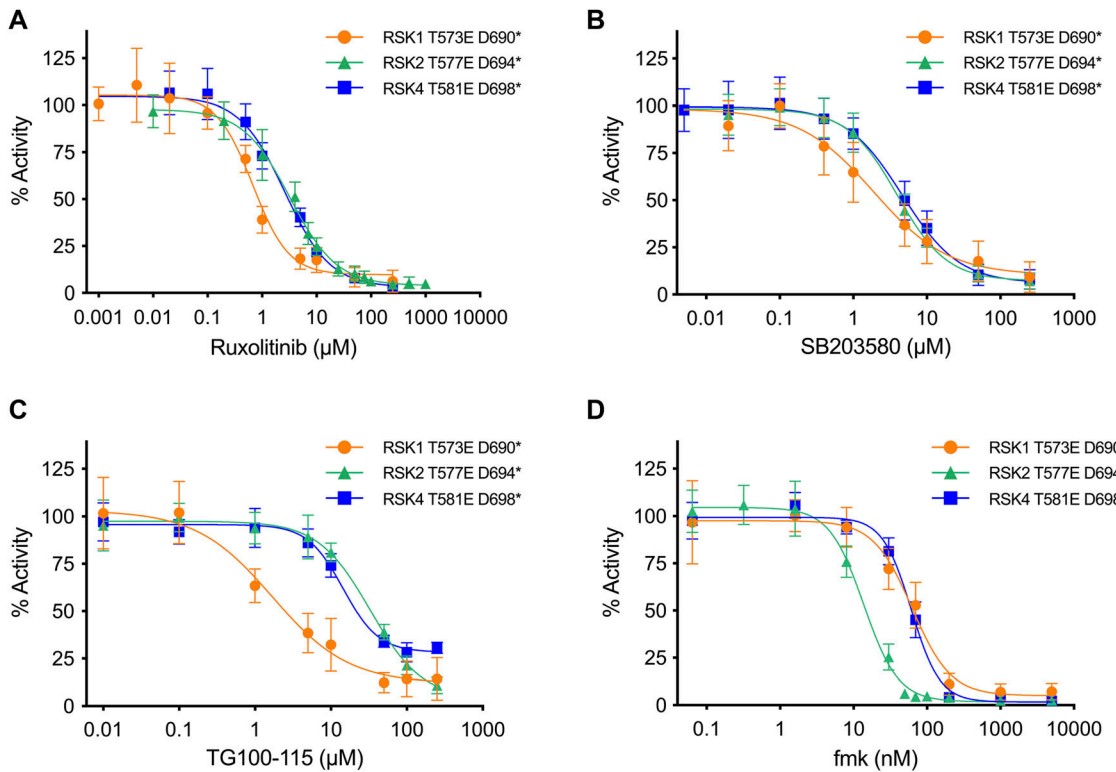

**Figure 4.  Dose-response inhibition curves.**
**(A, B, C, D)** Ruxolitinib, (B) SB203580, (C) TG-100-115, and (D) fmk for the three constitutively active mutants RSK1 T573E D690* (orange), RSK2 T577E D694* (green), and RSK4 T581E D698* (blue). All error bars indicate mean ± SD (n = 3, all data points included).

To date, only a small subset of RSK CTKD inhibitors has been described (16, 18, 21). To broaden the analysis, we tested the three most potent ATP-competitive inhibitors (ruxolitinib [INCB18424], SB-203580, and TG-100-115) that were identified earlier for the RSK1 and RSK4 CTKDs with $K_d$'s of 0.12–0.97 μM (15). The inhibitors displayed similar potency for all three RSK constructs with $IC_{50}$ values in the low micromolar range, except for TG-100-115 which was significantly more selective for the RSK1-T573E-D690* construct with a 19-fold reduction in $IC_{50}$ value compared with the equivalent RSK2-T577E-D694* (Table 2 and Fig 4). Furthermore, the well-known, highly specific, and irreversible RSK2 CTKD inhibitor, fmk (16) showed fourfold selectivity for the RSK2 construct with an $IC_{50}$

value of 13.7 nM for RSK2-T577E D694* compared with 61–63 nM for RSK1 and RSK4 (Table 2 and Fig 4).

Finally, the constitutively active mutant of the RSK2 CTKD enabled us also to evaluate fmk potency by the specificity constant ($k_{inact}/K_i$), which is a preferred measure for potency ranking of covalent irreversible inhibitors (22, 23, 24, 25). Covalent inhibition proceeds in two steps: first, the inhibitor associates reversibly to the protein, and second, a covalent bond is formed that results in the inactivated protein–inhibitor complex (Fig 5A). In this two-step mechanism, $K_i$ refers to the noncovalent binding constant (affinity of the ligand for the enzyme) and $k_{inact}$ to the rate constant for enzyme inactivation (24, 25). A kinase reaction buffer with lowered

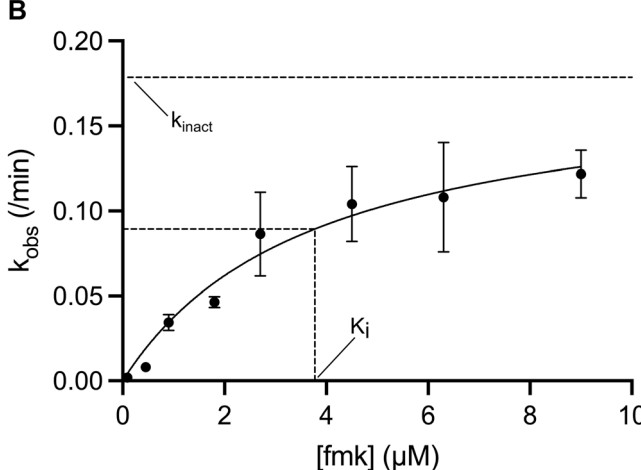

**Figure 5. Covalent inhibition.**
**(A)** Two-step mechanism of inhibition, where E reflects the enzyme of interest and I is the inhibitor. In the first step, the inhibitor binds reversibly to the enzyme creating the E–I complex, followed by a covalent bond formation in the inactivated E–I complex. The potency of the first reversible reaction is defined by $K_i$, and the potential maximal rate of inactivation is defined by $k_{inact}$. **(B)** Kinetic plot ($k_{obs}$ versus time-dependent inhibitor concentration) illustrating time-dependent loss of RSK2 T577E D694* activity with fmk, resulting in $k_{inact}/K_i$ constant of 790 ± 281 $M^{-1}$ $s^{-1}$, $k_{inact}$ = 0.18 ± 0.03 $min^{-1}$, and $K_i$ = 3.8 ± 1.2 $\mu M$ (n = 3). All error bars indicate mean ± SD.

pH (pH 6) was applied to slow down the reaction. A hyperbolic dependence of the observed rate constants, $k_{obs}$, on fmk concentration was observed, resulting in a $k_{inact}/K_i$ constant of 790 ± 281 $M^{-1}$ $s^{-1}$ at pH 6 with a maximum rate constant for inactivation $k_{inact}$ of 0.18 ± 0.03 $min^{-1}$, and an apparent inhibition constant $K_i$ of 3.8 ± 1.2 $\mu M$ (Fig 5B).

## Discussion

The RSK CTKDs are activated by threonine phosphorylation by the upstream kinase ERK1/2. When creating a phosphorylation mimic mutant (threonine to glutamate, RSK2 T577E), only a small increase of activity was obtained (as reported for RSK1 (26)), illustrating the presence of other autoregulatory elements and we therefore focused on the autoinhibitory regulation by the αL helix. To gain more insight into how αL regulates the RSK CTKDs, a series of RSK2-T577E mutants with single point mutations on αL was constructed and analyzed in vitro. In previous BHK-21 cell-based studies by Poteet-Smith et al (13), Y707A mutation of RSK2 showed moderate auto-phosphorylation at S386 (CTKD activity) in non-stimulated cells. Upon EGF stimulation, a high phosphorylation level similar to wt RSK2 was obtained (fivefold increase). In contrast, the Y707A mutant version with the threonine phosphorylation mimic (RSK2-T577E-Y707A) showed only a low 6% activity, comparable to the RSK2-T577E mutant alone (Fig 2D). This indicated that the disruption of the

hydrogen bond between S603 (αF) and Y707 (αL) alone is not sufficient to release the αL helix. From the structure of the inactive RSK2 CTKD (12) it was hypothesized that the main inhibitory role of the helix is defined by the formation of an ionic bond between E500 (αD) and K700 (αL). However, both the RSK2-T577E-K700A and the αL double mutant (RSK2-T577E-K700A-Y707A) also showed no significant additional gain of activity compared with the RSK2-T577E and RSK2-T577E-Y707A constructs. However, a high gain of constitutive activity (66% of activated wt RSK2 CTKD) was obtained by truncation at the N-terminal part of the αL helix on the RSK2-T577E mutant (RSK2-T577E-D694*), at level with an in vivo study (13). Similarly, the constitutive activity for the corresponding RSK1 and RSK4 CTKD constructs (RSK1-T573E-D690* and RSK4-T580E-D698*) demonstrated a general applicability of deletion of αL helix inhibition (Fig 2E)—and an autoregulatory mechanism of the αL helix is also observed in the structure of the inactive RSK1 CTKD (14).

This suggests that for the activated RSK/MSK kinases, the autoinhibitory αL helix adopts a specific active state conformation, either stabilized or displaced by ERK2 interaction. In the structure of the inactive RSK1–ERK2 complex, the Ala-Pro-Glu motif (APE motif, conserved segment of the kinase activation loop/P loop involved in ATP cofactor binding) in the C-terminal end of the extended αF helix of RSKs was found to play a dual role (see Fig 6A) (27). In the inactive RSK1 state, the APE motif was observed to be important for activator kinase (ERK2) binding, ensuring alignment of the kinases before phosphorylation of the RSK1 activation loop by ERK2. In the active state, this motif now plays a pivotal role in the RSK substrate binding in the catalytic cleft (28). Based on the RSK1–ERK2 complex structure and the structure of the inactive RSK2 CTKD (12), it was therefore proposed that the APE motif region undergoes a major conformational change that is triggered by phosphorylation of the activation loop of RSK CTKD by the ERK activator kinase, and that this motif presumably causes the αL displacement (27).

In our study, the activation loop of the RSK2 CTKD construct is affected by a phosphomimetic mutation (T577E), but this was not sufficient to activate RSK2, as has also been noted by Somale et al (31). Even when alanine point mutations were introduced in the αL helix to further destabilize interactions with αD and αF, the autoinhibition was not relieved in the absence of ERK2 interaction. This suggests that activation of the RSK CTKD depends on proper arrangement of the phosphorylated activation loop next to the substrate-binding cleft, and that this arrangement may be facilitated by ERK2 on its way to reach additional target sites (T363 and S369) in the linker region between NTKD and CTKD. The conformational change then moves the αL helix out of the way, eventually creating an active RSK CTKD (Fig 6). Hence, to engineer a constitutive activity of RSK2 CTKD, we truncated the αL helix to circumvent the need of its displacement by an effector kinase interaction. It is worth noting that truncation of the αL helix and phosphomimetic mutation may affect the RSK CTKD sensitivity and specificity toward the substrate, as observed with the high gain of activity for the RSK4-T580E-D698* mutant (Fig 2E).

In the serum-deprived HEK293 cells, a robust activation of human RSK2 T577E D694* protein was observed, as to be expected from the in vitro results (Fig 3). For the CREB phosphorylation, previous studies have indicated that under cellular stresses, the related MSK1/2 become the pre-dominant CREB kinases (rather than the

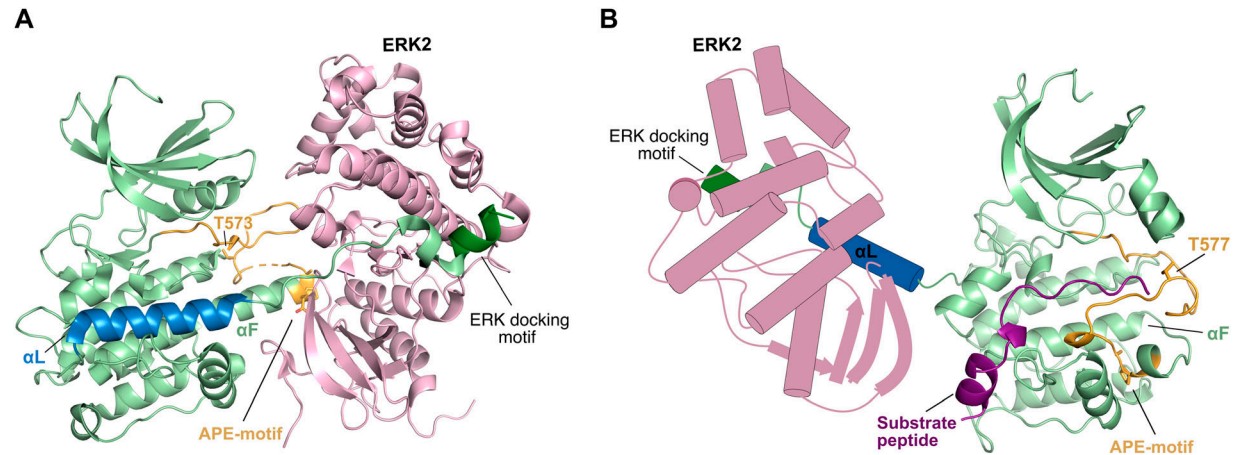

**Figure 6. Activation of the RSK CTKD.**
**(A)** Cartoon illustration of the structure of the inactive RSK1–ERK2 complex (PDB 4NIF (27)). The kinases are depicted using the following color scheme; green (RSK1 CTKD), blue (αL helix), orange (activation loop), dark green (ERK docking motif), and pink (ERK2). APE motif and T573 are shown as orange sticks. ERK interacts with RSK1 both at the docking motif and at the APE-motif, which ensure proper alignment of the kinases for phosphorylation of the T573 in the RSK1 activation loop. **(B)** Homology model of activated RSK2 CTKD in complex with substrate peptide (shown in deep purple) made with modeller (29) and based on the active complex of protein kinase A (PDB 2CPK (30), chain A). ERK2 and the αL helix are shown as schematic models. Activation of RSK CTKD depends on proper arrangement of the activation loop, likely guided by ERK2, and its movement is triggered by T577 phosphorylation. Correct alignment of the activation loop moves the αL helix out of the catalytic cleft for substrate phosphorylation, eventually creating an active RSK CTKD.

RSKs) (6, 32). In addition, from a study by Pirkmajer et al (33), serum starvation of HEK293 cells was shown to have an increased phosphorylation level of Akt, another well-known upstream kinase of YB-1 (33, 34).

The constitutively active mutants of the RSK CTKDs gave us the ability to study CTKD activity without the interference of an ERK pre-phosphorylation procedure. This enabled us to determine apparent $K_M$(ATP) for the RSK CTKDs, resulting in values in the range 40–107 $\mu$M. To our knowledge, this constant has not previously been determined. In comparison, the related MK2 kinase with a similar autoinhibitory helix deletion showed a $K_M$(ATP) of 15 $\mu$M (35). Furthermore, values in the same range (3–120 $\mu$M) have been reported for the HTRF KinEASE assay for other members of the MAPKAPKs belonging to the CaMK family (e.g., MNK1/2 and MK2/3) ((1), HTRF KinEASE, Cisbio), so all in similar ranges to the values we report here for RSK CTKD constructs.

The three ATP-competitive inhibitors (ruxolitinib [INCB18424], SB-203580, and TG-100-115) showed moderate potency in the low micromolar range to our constitutively active RSK CTKD constructs. They are somewhat selective toward other kinases. Overall, the RSK1 construct showed higher sensitivity toward the tested inhibitors, and in general all RSKs showed highest sensitivity to ruxolitinib, followed by TG-100-115 for RSK1 and SB203580 for RSK2 and RSK4. The same tendency was concluded by Davis et al (15). Compared with the inhibitors' general kinase target profiles, ruxolitinib, considered a highly potent JAK2 inhibitor ($IC_{50}$ = 3.3 nM) (36), showed a very different $IC_{50}$-value for RSK2 T577E D694* ($IC_{50}$ = 3.2 $\mu$M, i.e., ~1,000-fold higher). Oppositely, SB203580 (p38 protein inhibitor, $IC_{50}$ = 80 nM (37)) showed $IC_{50}$ values of RSK inhibition that are only 20- to 60-fold higher. Binding of SB203580 to p38 protein depends on a threonine gatekeeper in the ATP-binding pocket. The RSK CTKDs contain an equivalent threonine (e.g., Thr493 for RSK2) which may explain why this intended p38-specific inhibitor shows

quite comparable $IC_{50}$-values to RSKs. Intermediate of these, the PI3Kγ selective inhibitor TG-110-115 ($IC_{50}$ = 83 nM (38)) showed $IC_{50}$-values that are 20- to 400-fold higher for the constitutively active RSK CTKD constructs.

Approximative $IC_{50}$ values for the covalent fmk inhibitor to the three RSK constructs were observed to be in the same range as for wt RSK2 CTKD ($IC_{50}$ of 15 nM) (16) (Table 2). However, fmk is a covalent inhibitor, and its inhibitory reaction is in practice irreversible and will gradually proceed to complete neutralization of the protein target rather than showing equilibrium kinetics (22). The specificity constant ($k_{inact}/K_i$) for a two-step binding and bond formation reaction is the appropriate potency indicator for covalent inhibitors (22, 23, 24, 25). RSK2 T577E D694* allowed this time-dependent inhibition study to be conducted for fmk at pH 6, which resulted in a moderate potency with $k_{inact}/K_i$ = 790 ± 281 $M^{-1}$ $s^{-1}$. At pH 7.4, the rate of inactivation ($k_{obs}$) observed at higher inhibitor concentrations tested (2.7–9.0 $\mu$M) was too fast for the assay used here, and the reaction proceeded to complete neutralization within few seconds (data not included). At pH 6, the slower kinetics result from a reaction of a primarily protonated Cys436 thiol, which is significantly less reactive than the thiolate form (indicated by the low $k_{inact}$ value). In addition, the altered protonation state (chemical environment) at pH 6 may induce conformational changes or steric effects near the active site hindering a tight binding of the fmk inhibitor to Cys436, as implied by the low $K_i$ value (3.8 ± 1.2 $\mu$M). If indeed the case, $K_i$ would expectedly be in the lower nanomolar range (similar to $IC_{50}$) at neutral pH, resulting in a $k_{inact}/K_i$ value at the range of $10^3$–$10^6$ $M^{-1}$ $s^{-1}$ as known for other potent covalent kinase inhibitors targeting cysteines, including Bruton's tyrosine kinase (inhibitor 2; $k_{inact}$ = 3.5 × $10^{-3}$ $s^{-1}$, $K_i$ = 0.14 $\mu$M, $k_{inact}/K_i$ = 2.5 × $10^4$ $M^{-1}s^{-1}$ (39)), IL-2–inducible tyrosine kinase (compound 12; $k_{inact}$ = 2.6 × $10^{-3}$ $s^{-1}$, Ki = 5 nM, $k_{inact}/K_i$ = 5.2 × $10^5$ $M^{-1}s^{-1}$ (40)), and JAK3 (compound 7; $k_{inact}/K_i$ = 1.1 × $10^6$ $M^{-1}s^{-1}$ (41)). To elaborate on this

discussion, the effect of pH on the $k_{inact}/K_i$ should be explored and it could also provide an estimate of the $pK_a$ of the Cys436 and its potential to react with an electrophilic inhibitor (25).

Constitutively active RSK CTKD mutants could be powerful tools for screening large compound libraries as they allow for "ready-to-go" screening protocols. Circumventing a prior kinase activation step, the time-dependent covalent irreversible inhibitor studies ($k_{inact}/K_i$ determinations) can also be more easily performed, which is currently often missing information and important for inhibitor optimization (23). Along with the previously reported active mutant constructs of other αL-regulated CaMK family kinases (MK2 (35, 42, 43, 44) and CaMKI (45)), these constitutively active RSK mutants may thereby facilitate screening and development of novel kinase inhibitors, also with a covalent mode of action.

# Materials and Methods

## RSK kinases expression and purification from *E. coli*

The CTKD of human p90 RSK 1, 2, and 4 (RSK1, 2, and 4) with an N-terminal His-tag (His$_8$), a linker region (DYDIPTT), and a tobacco etch virus protease site (ENLYFQG) in a pET-22b vector were purchased from GenScript. Site-directed mutagenesis was performed using the QuikChange Lightning Site-Directed Mutagenesis Kit (Agilent Technologies). *E. coli* BL21 (DE3) Rosetta II was transformed with a RSK construct and plated on lysogeny broth (LB) agar plates containing 50 $\mu$g/ml ampicillin (Amp) and 35 $\mu$g/ml chloramphenicol (Cam). Five colonies were used to inoculate 20 ml of LB overnight culture containing 50 $\mu$g/ml Amp and 35 $\mu$g/ml Cam. 2 liters of LB, containing 100 $\mu$g/ml Amp and 35 $\mu$g/ml Cam, were inoculated with the overnight culture and grown at 37°C to $OD_{600}$ ~0.8. Expression was induced by the addition of IPTG. Thereafter, the temperature was lowered to 20°C for 3 h and further lowered to 12°C for 18 h. Cells were harvested by centrifugation and the cell pellets frozen at –80°C. For purification, cell pellets were thawed, re-suspended in 50 ml lysis buffer (50 mM Tris–HCl, pH 8.0, 100 mM NaCl, 10% [vol/vol] glycerol, 10 mM $\beta$-mercaptoethanol, and 2 mM PMSF), and lysed by sonication (3 × 5 min, 50% intensity, cooling in between). The cell debris was cleared by centrifugation at 25,000$g$ for 45 min and 5 ml of $Ni^{2+}$-beads slurry (Ni-sepharose 6 Fast Flow; GE Healthcare) were washed in washing buffer (20 mM Tris–HCl, pH 8.0, 100 mM NaCl, 10% [vol/vol] glycerol, 10 mM $\beta$-mercaptoethanol) prior incubation with supernatant for 1 h at room temperature. The $Ni^{2+}$-beads were washed with 100 ml washing buffer, and RSK was eluted in 2 × 5 ml elution buffer (20 mM Tris–HCl, pH 8.0, 100 mM NaCl, 10% [vol/vol] glycerol, 10 mM $\beta$-mercaptoethanol, and 500 mM imidazole). The eluate was supplemented with 1 mg of recombinant tobacco etch virus-protease and dialysed against 1 liter of washing buffer overnight at room temperature. Dialysed RSK was applied to the $Ni^{2+}$-beads and digested RSK was collected in the flow through. RSK was concentrated by centrifugation to 10 mg/ml (Vivaspin 20, 30 kD cutoff). Protein concentration was estimated by spectrophotometry (NanoDrop; Thermo Fisher Scientific) assuming $\varepsilon_{RSK2}$ = 44,350 $cm^{-1}$ and $M_{RSK2}$ = 38.4 kD (calculated for each construct).

## In vitro assay

In vitro activity of human RSK CTKD was evaluated in the HTRF KinEASE assay (Cisbio Bioassays). For activity evaluation, RSK CTKD proteins (2 $\mu$M) were either untreated or pre-activated by ERK2 (0.1 $\mu$M) (SignalChem) in reaction buffer containing 200 $\mu$M ATP and 10 mM $MgCl_2$ in kinase buffer (50 mM HEPES, pH 7.0, 2 mM TCEP, 0.02% [wt/vol] $NaN_3$, 0.01% [wt/vol] BSA, and 0.1 mM orthovanadate) for 1 h at 30°C. Reactions were performed in 384-well plates for fluorescence (Greiner) with 10 ng RSK CTKD per well. The reactions were started by the addition of 100 $\mu$M ATP and 1 $\mu$M STK1 substrate, giving a 10 $\mu$l reaction solution. After 20 min at room temperature, the reaction was stopped with 5 $\mu$l anti-phospho STK antibody labeled with $Eu^{3+}$-cryptate (FRET donor) and 5 $\mu$l streptavidin-XL665 (FRET acceptor), both prepared in EDTA. In inhibition studies, 10 ng constitutively active RSK CTKDs were mixed with DMSO or a concentration series of ruxolitinib, SB-203580, TG-100-115, or fmk in kinase buffer for a 30-min incubation prior kinase reaction (as described above). Evaluation of the time-dependent effect of fmk on RSK2-T577E-D694* was performed by incubation with a concentration series of fmk in kinase buffer (50 mM succinic acid, pH 6.0, 2.5 mM TCEP, 0.02% [wt/vol] $NaN_3$, 0.01% [wt/vol] BSA and 0.1 mM orthovanadate) for 1, 2, 4, 6 min or 2, 5, 15, 30 min (depending on concentration series) followed by kinase reaction. $IC_{50}$ values were calculated by non-linear regression using sigmoid concentration response. The observed rate constant of fmk inhibition, $k_{obs}$, at each concentration was determined from the slope of a semi-logarithmic plot of inhibition versus time. $k_{obs}$ was re-plotted against inhibitor concentration and fitted to a hyperbolic equation, $k_{obs} = k_{inact}[inhibitor]/(K_i + [inhibitor])$ (22). Data analysis was performed using GraphPad Prism 9. All experiments were repeated at least three times ($n$ = 3) and presented as means ± SD. The *P*-values were determined by *t* test.

## Phosphorylation analysis in human embryonic kidney cells

Phosphorylation analysis of full-length human RSK2 and human RSK2-T577E-D694 mutant was performed in the mammalian expression vector (pEBG2T), where constructs were fused with a N-terminal GST tag. The mutant construct was obtained by using the QuikChange Lightning Site-Directed Mutagenesis Kit (Agilent Technologies). Human embryonic kidney cells (HEK293) were cultured in a culture flask (150 $cm^2$) to 80% confluence in DMEM (Gibco) supplemented with 50 units/ml penicillin G (Gibco), 50 $\mu$g/ml streptomycin (Invitrogen), 5 $\mu$g/ml gentamycin (Gibco), and 10% (vol/vol) FBS (Gibco). Cells were trypsinated and seeded in six-well plates (9 $cm^2$) in 2 ml DMEM supplemented with 10% (vol/vol) FBS and antibiotics (penicillin G, streptomycin, and gentamycin), and incubated for 3–4 d until 80% confluence. The medium was changed to DMEM with antibiotics for 16 h. The cells were transfected using 440 ng plasmid DNA dissolved in 31 $\mu$l DMEM (+antibiotics) and 4 $\mu$l lipofectamine dissolved in 27 $\mu$l DMEM (+antibiotics) added together for 20 min before transferred to the cells. The cells were incubated with the lipofectamine/DNA plasmid mixture for 4 h at 37 and 5% $CO_2$. Cells at time point 0 were harvested after these 4 h. For the remaining cells, the medium was changed to DMEM with antibiotics, but without FBS to hinder stimulation of the cells (e.g., stimulation MAPK-signaling pathway). These cells were harvested after 6, 15, 24, and 48 h,respectively, by washing

with PBS and subsequent addition of 50 µl lysis buffer supplemented with 1 mM PMSF/ml buffer and 22 µl/ml buffer protease inhibitor cocktail (Complete, Roche Diagnostics). The cell samples were boiled for 3 min, followed by quick centrifugation, addition of 0.5 µl of benzoase (Merck), and incubating on ice for 30 min before protein determination. 40 µg of protein were separated on 4–15% Mini-Protean TGX Pre-cast SDS–PAGE % gels (Bio-Rad). Proteins were blotted onto Trans-Blot Turbo Transfer system nitrocellulose membranes (Bio-Rad) and tested with antibodies: anti-phospho-p90RSK (S380) #9341, anti-phospho-RSK2 (S227) #3556, anti-RSK1/RSK2/RSK3 #9355, anti-$\beta$-actin #8457, and HRP anti-rabbit #7074, (all from Cell Signaling). Gels were cut horizontally below the 65 kD mark to treat higher and lower molecular weight probes on separate blots. Membranes were evaluated using a C-DiGit Blot scanner (LI-COR Biosciences).

## Supplementary Information

## Acknowledgments

We are grateful to technicians Anna Marie Nielsen and Annette Blak Rasmussen (Aarhus University) for technical assistance. We wish to thank Professor J Simon C Arthur (University of Dundee) for providing the pEBG2T plasmid. JL Andersen, L Iversen, and P Nissen were supported by a pre-seed grant from the Novo Nordisk Foundation (grant no. NNF13SA0006011). P Nissen was supported through the DANDRITE center and a professorship financed by the Lundbeck Foundation (grant no. R248-2016-2518 and R310-2018-3713). MU Fruergaard was supported by a PhD stipend cofinanced by the Graduate School of Natural Sciences, Aarhus University.

### Author Contributions

MU Fruergaard: formal analysis, validation, investigation, visualization, and writing—original draft, review, and editing.
CJF Nielsen: formal analysis, validation, and investigation.
CR Kjeldsen: formal analysis, validation, and investigation.
L Iversen: resources, data curation, formal analysis, supervision, funding acquisition, validation, methodology, and project administration.
JL Andersen: conceptualization, resources, formal analysis, supervision, funding acquisition, validation, investigation, methodology, and project administration.
P Nissen: conceptualization, resources, data curation, formal analysis, supervision, funding acquisition, validation, investigation, visualization, and writing—original draft, review, and editing.

### Conflict of Interest Statement

The authors declare that they have no conflict of interest.

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
