## [Reviewer comments · Life Science Alliance]

Life Science Alliance

Activation and inhibition of the C-terminal kinase domain of p90 ribosomal S6 kinases

Marlene Fruergaard, Christine Nielsen, Cecilia Rosada Kjeldsen, Lars Iversen, Jacob Andersen, and Poul Nissen
DOI: <https://doi.org/10.26508/lsa.202201425>

Corresponding author(s): Poul Nissen, Aarhus University

Review Timeline:	Submission Date:	2022-02-25
	Editorial Decision:	2022-03-29
	Revision Received:	2023-01-22
	Editorial Decision:	2023-01-26
	Revision Received:	2023-02-08
	Accepted:	2023-02-09

Scientific Editor: Novella Guidi

Transaction Report:

March 29, 2022

Re: Life Science Alliance manuscript #LSA-2022-01425-T

Prof. Poul Nissen
Aarhus University
Molecular Biology and Genetics
Gustav Wieds Vej 10C
Aarhus 8000
Denmark

Dear Dr. Nissen,

Thank you for submitting your manuscript entitled "Activation and inhibition of the C-terminal kinase domain of p90 ribosomal S6 kinases" to Life Science Alliance. The manuscript was assessed by expert reviewers, whose comments are appended to this letter. We, thus, encourage you to submit a revised version of the manuscript back to LSA that responds to all of the reviewers' points.

Thank you for this interesting contribution to Life Science Alliance. We are looking forward to receiving your revised manuscript.

Sincerely,

B. MANUSCRIPT ORGANIZATION AND FORMATTING:

Reviewer #1 (Comments to the Authors (Required)):

In this paper Marlene Uglebjerg Fruergaard and colleagues aim at developing a constitutive active version of the C terminal kinase of p90 ribosomal S6 kinases (RSKs). This was a problem that has stayed unsolved in the scientific community so far. A complete understanding of the complex regulation of these dual kinase proteins is still not accomplished. However, this study finally manages to produce a true constitutively active mutant of the C terminal kinase. This achievement is a significant step forward in the understanding of such a regulatory mechanism and it will be extremely useful for future studies. The authors identified the constitutively active mutant, that is composed by a phosphomimetic mutation plus a truncation, by mutational studies, in vitro kinase activity assay, experiments in cells and dose-response to 4 small molecule inhibitors. I consider this manuscript appropriate for Life Science Alliance in case the authors decide to improve the manuscript according to my suggestions.

Major concerns:

- 1) Figure 3 is missing the control blot to check the expression of the exogenous construct. For instance, it would be important to show an immunoblot anti-GST.
- 2) A second critical point related to the previous one is the normalization strategy that the authors adopted to compare RSK2 wild-type and the mutant in immunoblot of the phosphorylated S386 and S227 in cells. The author claim that "The signals were normalized to the endogenous RSK" to produce the charts in 3B and C, but for me this approach is wrong. First, why normalizing with the endogenous given that the endogenous and exogenous are clearly distinguishable by size? Second, it doesn't take in account possible differences between the expression levels of the exogenous RSK2 wild-type and the mutant. Third, it is impossible to claim that there is a difference in phosphorylation between RSK2 wild-type and the mutant unless they are expressed at the exact same amount (fact that is not proven in the current version of the paper). To solve this problem the authors should normalize the phosphorylated protein against the total amount of expressed exogenous protein. It can be an anti-GST antibody or also an anti-total RSK2.
- 3) Legend of figure 2 is completely messed up, There is two times "A)". D and E are not existing.

Minor concerns:

- 1) It is unclear what the authors mean with "regulate many cellular processes, such as cell regulation" in the introduction. It sounds to me a circular logic.
- 2) In the introduction BI-DL1870 should be without "L".
- 3) At page 8, "The inhibitors displayed similar potency for all three RSK constructs with IC50 values in the low micromolar range, except for TG-100-115 which was significantly more selective for the RSK1-T573E-D690* construct with a 19-fold reduction in IC50 value compared to the equivalent RSK2-T577E-D694* (table 2, figure 5)" should be linked to figure 4 not 5.
- 4) Figures 2D, 2E, 3B, 3C and 4A-D would benefit from a small legend in the figure showing the meaning of the various colors.

Reviewer #2 (Comments to the Authors (Required)):

This paper is concise and to the point description of creating specific mutations that result to constitutively active CDK kinase domain of RSK2. The structure guided mutagenesis experiments, leading to the combination if a phospho-memetic mutant in

combination with a C-terminal truncation, is well-argued and clearly presented. The resulting mutants are a useful tool for characterising inhibitors, and this is exemplified robustly with in vitro experiments and in cells.

All data presented for all in vitro experiments are very clear.

If anything, the authors might want to consider - albeit I do not find it crucial - doing the in situ experiment in an additional different cell line (e.g. fibroblast?) to ensure that their finding is not affected by the idiosyncrasies of HEK293 cells.

A minor comment: "Worth noting, the mutant harboring only the truncation (RSK2-D694*) displayed only significant activity upon ERK activation. " I think the authors mean "Worth noting, the mutant harboring only the truncation (RSK2-D694*) displayed significant activity only upon ERK activation."

And a preference: while e.g. T577E is fine, I prefer Thr577 when referring to the residues in text.

Reviewer #3 (Comments to the Authors (Required)):

This study is nicely presented, contains substantial amount of biochemical data, but it has a limited scope. The authors generated a constitutively active version of the C-terminal kinase domain of RSK (CTKD). This construct harbors a phospho-mimicking mutation in the activation loop (AL) and has the C-terminal inhibitory helix (alphaL) truncated. The authors claim that this construct will be useful in order to find new drugs because the screens could be carried out easier since there is no need to add the activating ERK kinase into the kinase reaction mix. They demonstrate this point by determining the IC50 of four known RSK inhibitors, where one of them was a covalent inhibitor (fmk). In addition, they also determine the KM for ATP, which fall within the expected high micromolar range for RSK1, RSK2 and RSK4. The fact that the authors obtained the constitutively active CTKD by combining an AL mutation and alphaL truncation is not surprising based on former data/knowledge. Overall, the data is good and it is nicely presented, but the study is only descriptive and provides little mechanistic insight. The manuscript is well-written and it is suitable for publication after providing some more discussion of the data (see below).

Minor comments:

1) I think the results shown on Figure 2B need more discussion. This data shows that the active CTKD has substrate preference, but the nature of this specificity is not discussed. What was the difference between STK1, STK2, and STK3?

2) The data shown on Figure 5B would need some more discussion. The authors state that the irreversible fmk inhibitor showed a moderate potency at pH 6 (790 1/M.s). What is the significance of this finding? I think this number and the fact that the experiment was done at a lowered pH so that to slow down the reaction need to be put into perspective (e.g. what would one expect at a higher, more physiological pH? what value would qualify for a high potency inhibitor in comparison with other known examples?). Do the authors intend to imply that it is the low micromolar (3.8 microM) Ki that makes fmk a moderate inhibitor or its slow kinetic rate (0.18 1/min) (albeit these were measured at pH 6 so that to slow down the latter). Would not this type of analysis warrant measurements under different pH-s?

Reviewer comment

Our response

Reviewer #1 (Comments to the Authors (Required)):

In this paper Marlene Uglebjerg Fruergaard and colleagues aim at developing constitutive active version of the C terminal kinase of p90 ribosomal S6 kinases (RSKs). This was a problem that has stayed unsolved in the scientific community so far. A complete understanding of the complex regulation of these dual kinase proteins is still not accomplished. However, this study finally manages to produce a true constitutively active mutant of the C terminal kinase. This achievement is a significant step forward in the understanding of such a regulatory mechanism and it will be extremely useful for future studies. The authors identified the constitutively active mutant, that is composed by a phosphomimetic mutation plus a truncation, by mutational studies, in vitro kinase activity assay, experiments in cells and dose-response to 4 small molecule inhibitors.

I consider this manuscript appropriate for Life Science Alliance in case the authors decide to improve the manuscript according to my suggestions.

Major concerns:

1) Figure 3 is missing the control blot to check the expression of the exogenous construct. For instance, it would be important to show an immunoblot anti-GST.

Response: We agree. In the revised figure 3 anti-total RSK2 has been included to quantify the total amount of exogenous RSK2 expressed and relate the phosphorylated RSK2 to that.

2) A second critical point related to the previous one is the normalization strategy that the authors adopted to compare RSK2 wild-type and the mutant in immunoblot of the phosphorylated S386 and S227 in cells. The author claim that "The signals were normalized to the endogenous RSK" to produce the charts in 3B and C, but for me this approach is wrong.

First, why normalizing with the endogenous given that the endogenous and exogenous are clearly distinguishable by size? Second, it doesn't take in account possible differences between the expression levels of the exogenous RSK2 wild-type and the mutant. Third, it is impossible to claim that there is a difference in phosphorylation between RSK2 wild-type and the mutant unless they are expressed at the exact same amount (fact that is not proven in the current version of the paper).

To solve this problem the authors should normalize the phosphorylated protein against the total amount of expressed exogenous protein. It can be an anti-GST antibody or also an anti-total RSK2.

Response: We agree. The relative abundance of phosphorylated RSK2 proteins (in revised figure 3) have been normalized to the total exogenous RSK2 levels.

3) Legend of figure 2 is completely messed up, There is two times "A)". D and E are not existing.

Response: We apologize for the oversight. The panel numbers are now corrected.

Minor concerns:

1) It is unclear what the authors mean with "regulate many cellular processes, such as cell regulation" in the introduction. It sounds to me a circular logic.

Response: The phrase "cell regulation" has been omitted in the revised manuscript.

2) In the introduction BI-DL1870 should be without "L".

Response: BI-DL1870 is now corrected to BI-D1870

3) At page 8, "The inhibitors displayed similar potency for all three RSK constructs with IC50 values in the low micromolar range, except for TG-100-115 which was significantly more selective for the RSK1-T573E-D690* construct with a 19-fold reduction in IC50 value compared to the equivalent RSK2-T577E-D694* (table 2, figure 5)" should be linked to figure 4 not 5.

Response: The figure reference is now corrected to figure 4.

4) Figures 2D, 2E, 3B, 3C and 4A-D would benefit from a small legend in the figure showing the meaning of the various colors.

Response: Color legends are now included in figure 2, 3 and 4.

Reviewer #2 (Comments to the Authors (Required)):

This paper is concise and to the point description of creating specific mutations that result to constitutively active CDK kinase domain of RSK2. The structure guided mutagenesis experiments, leading to the combination of a phospho-mimetic mutant in combination with a C-terminal truncation, is well-argued and clearly presented. The resulting mutants are a useful tool for characterising inhibitors, and this is exemplified robustly with in vitro experiments and in cells.

All data presented for all in vitro experiments are very clear.

If anything, the authors might want to consider - albeit I do not find it crucial - doing the *in situ* experiment in an additional different cell line (e.g. fibroblast?) to ensure that their finding is not affected by the idiosyncrasies of HEK293 cells.

Response: We appreciate the suggestion, but we think that a gain of activity for a mutated form is shown well from the combination of *in vitro* experiments and a single cell line experiment. We would certainly agree to the possible pitfalls of e.g. HEK293 idiosyncrasy if we were describing a novel mechanism or a loss of function.

A minor comment: "Worth noting, the mutant harboring only the truncation (RSK2-D694*) displayed only significant activity upon ERK activation. " I think the authors mean "Worth noting, the mutant harboring only the truncation (RSK2-D694*) displayed significant activity only upon ERK activation."

Response: Indeed, the sentence has been corrected as suggested.

And a preference: while e.g. T577E is fine, I prefer Thr577 when referring to the residues in text.

Response: To ensure consistency and readability, we have used one-letter codes throughout.

Reviewer #3 (Comments to the Authors (Required)):

This study is nicely presented, contains substantial amount of biochemical data, but it has a limited scope. The authors generated a constitutively active version of the C-terminal kinase domain of RSK (CTKD). This construct harbors a phospho-mimicking mutation in the activation loop (AL) and has the C-terminal inhibitory helix (alphaL) truncated. The authors claim that this construct will be useful in order to find new drugs because the screens could be carried out easier since there is no need to add the activating ERK kinase into the kinase reaction mix. They demonstrate this point by determining the IC50 of four known RSK inhibitors, where one of them was a covalent inhibitor (fmk). In addition, they also determine the KM for ATP, which fall within the expected high micromolar range for RSK1, RSK2 and RSK4. The fact that the authors obtained the constitutively active CTKD by combining an AL mutation and alphaL truncation is not surprising based on former data/knowledge. Overall, the data is good and it is nicely presented, but the study is only descriptive and provides little mechanistic insight. The manuscript is well-written and it is suitable for publication after providing some more discussion of the data (see below).

Minor comments:

1) I think the results shown on Figure 2B need more discussion. This data shows that the

active CTDK has substrate preference, but the nature of this specificity is not discussed. What was the difference between STK1, STK2, and STK3?

Response: The sequences of the STK substrates are unfortunately not disclosed with the HTRF KinEASE kit (Cisbio). The specificity therefore cannot be discussed here.

2) The data shown on Figure 5B would need some more discussion. The authors state that the irreversible fmk inhibitor showed a moderate potency at pH 6 (790 1/M.s). What is the significance of this finding? I think this number and the fact that the experiment was done at a lowered pH so that to slow down the reaction need to be put into perspective (e.g. what would one expect at a higher, more physiological pH? what value would qualify for a high potency inhibitor in comparison with other known examples?). Do the authors intend to imply that it is the low micromolar (3.8 microM) K_i that makes fmk a moderate inhibitor or its slow kinetic rate (0.18 1/min) (albeit these were measured at pH 6 so that to slow down the latter). Would not this type of analysis warrant measurements under different pH-s?

Response: a more elaborate discussion on the pH effect on the potency of fmk is now included. The expected k_{inact}/K_i value at physiological pH is compared to other known covalent inhibitors targeting cysteine residues. The kinetic assay used here had 2 seconds as the shortest time point possible for robust, reproducible results. As the reactions led to complete neutralization/inhibition within few seconds of reaction at higher inhibitor concentrations at pH 7.4, the rate of inactivation (k_{obs}) could not be determined directly by the assay applied. Therefore, the pH dependence cannot be examined well from this study, but it could for sure be expanded in a dedicated study of such kinetics.

Rather, the k_{inact}/K_i value obtained in this manuscript serves as a “proof-of-concept” for the applicability of the designed, constitutively active RSK2 T577E D694* in covalent inhibitor screening protocols for RSK2, and we discuss that in experiments dedicated to the development or characterization of such inhibitors, the construct design presented here will be useful.

January 26, 2023

RE: Life Science Alliance Manuscript #LSA-2022-01425-TR

Prof. Poul Nissen
Aarhus University
Molecular Biology and Genetics
Universitetsbyen 81, bld. 1874
Aarhus C 8000
Denmark

Dear Dr. Nissen,

Thank you for submitting your revised manuscript entitled "Activation and inhibition of the C-terminal kinase domain of p90 ribosomal S6 kinases". We would be happy to publish your paper in Life Science Alliance pending final revisions necessary to meet our formatting guidelines.

- please address the final Reviewer 3's comment
- please add the author contributions to the main manuscript text
- please use the [10 author names, et al.] format in your references (i.e. limit the author names to the first 10)
- please add a figure callout for Figure 2A; Figure 3 A,B; and Figure 6B; please note that if you add a callout for individual panels of a figure, you need to add a callout for each panel

A. FINAL FILES:

B. MANUSCRIPT ORGANIZATION AND FORMATTING:

Sincerely,

Reviewer #3 (Comments to the Authors (Required)):

The authors revised the manuscript. The new text written on the relevance of k_{inact}/K_i value determination at a non-physiological value is satisfactory. However, the authors should correct a factual mistake in the following sentence in what I encountered in the new text:

"If indeed the case, K_i would expectedly be in the lower nanomolar range (similar to IC_{50}) at neutral pH resulting in a k_{inact}/K_i value at the range of 10^{-3} - 10^{-6} $M^{-1} s^{-1}$ as known for other potent covalent kinase inhibitors targeting cysteines"

The typical k_{inact}/K_i values range, in the specified unit ($M^{-1} s^{-1}$) is not what is shown next in the listed concrete examples for BTK, ITK and JAK3. There the values are from 10^4 to 10^6 $M^{-1} s^{-1}$, so correctly a typical range is rather between 10^3 - 10^6 $M^{-1} s^{-1}$ for k_{inact}/K_i .

February 8, 2023

RE: Life Science Alliance Manuscript #LSA-2022-01425-TRR

Prof. Poul Nissen
Aarhus University
Molecular Biology and Genetics
Universitetsbyen 81, bld. 1874
Aarhus C 8000
Denmark

Dear Dr. Nissen,

Thank you for submitting your Research Article entitled "Activation and inhibition of the C-terminal kinase domain of p90 ribosomal S6 kinases". It is a pleasure to let you know that your manuscript is now accepted for publication in Life Science Alliance. Congratulations on this interesting work.

DISTRIBUTION OF MATERIALS:

Again, congratulations on a very nice paper. I hope you found the review process to be constructive and are pleased with how the manuscript was handled editorially. We look forward to future exciting submissions from your lab.

Sincerely,
